# New Growth-Related Features of Wheat Grain Pericarp Revealed by Synchrotron-Based X-ray Micro-Tomography and 3D Reconstruction

**DOI:** 10.3390/plants12051038

**Published:** 2023-02-24

**Authors:** David Legland, Thang Duong Quoc Le, Camille Alvarado, Christine Girousse, Anne-Laure Chateigner-Boutin

**Affiliations:** 1INRAE, UR BIA, 44316 Nantes, France; 2INRAE, PROBE Research Infrastructure, BIBS Facility, 44316 Nantes, France; 3INRAE, Université Clermont-Auvergne, UMR GDEC, 63000 Clermont-Ferrand, France

**Keywords:** grain development, 3D imaging, spatial variability, pericarp, mesocarp, cross cells, tube cells, stomata

## Abstract

Wheat (*Triticum aestivum* L.) is one of the most important crops as it provides 20% of calories and proteins to the human population. To overcome the increasing demand in wheat grain production, there is a need for a higher grain yield, and this can be achieved in particular through an increase in the grain weight. Moreover, grain shape is an important trait regarding the milling performance. Both the final grain weight and shape would benefit from a comprehensive knowledge of the morphological and anatomical determinism of wheat grain growth. Synchrotron-based phase-contrast X-ray microtomography (X-ray µCT) was used to study the 3D anatomy of the growing wheat grain during the first developmental stages. Coupled with 3D reconstruction, this method revealed changes in the grain shape and new cellular features. The study focused on a particular tissue, the pericarp, which has been hypothesized to be involved in the control of grain development. We showed considerable spatio-temporal diversity in cell shape and orientations, and in tissue porosity associated with stomata detection. These results highlight the growth-related features rarely studied in cereal grains, which may contribute significantly to the final grain weight and shape.

## 1. Introduction

Worldwide, wheat is a major crop used mainly for food and animal feed, with a central role in the human diet providing a substantial part of daily required calories, protein, and other pro-health compounds [1]. To meet the rising demand for wheat production, one solution is to increase the final weight of the individual grain [2]. Moreover, from a technological point of view, the shape itself is important as it influences the milling performance, an important step of grain processing [3]. The final grain weight and shape of wheat grains result from complex and coordinated transformations of the grain dimensions over the entire course of its development. To increase our understanding of these transformations, it is essential—as a first step—to gain a more precise description of changes in the morphology and anatomical properties of the cells and tissues that form the grain.

The final grain shape and size result from developmental processes occurring in various compartments of the grain. Botanically, the wheat grain is a fruit and, more precisely, a caryopsis containing a single true seed defined as a matured ovule. In this article, we will refer to this definition of the seed and not to its broader definition as a unit of dissemination. A mature wheat caryopsis is composed of the seed containing the embryo, a storage tissue, the starchy endosperm itself surrounded by a layer of specialized endosperm cells, the aleurone layer, and several compressed outer cell layers, including the remains of the seed nucellar epidermis and testa, and that of the pericarp [4]. The caryopsis development, which is initiated by the ovule fertilization, is classically divided into the (i) early (lag-) phase, (ii) the filling phase, mainly characterized by a rapid accumulation of dry matter and by a water plateau, and (iii) the maturation phase characterized by the caryopsis dehydration and polymerization of storage proteins.

The development of cereal endosperm and embryo has been extensively studied [4,5,6,7,8,9]. On the contrary, the development of the outer layers (pericarp + testa and nucellar epidermis) of the cereal caryopsis has been less comprehensively studied. Nevertheless, pericarp was suggested to greatly contribute to the determinism of the wheat caryopsis size, weight, and shape [10]. Indeed, at early stages, the caryopsis consists mainly of pericarp cells that extend, causing grain elongation. During this early development, the wheat grain shape changes from a triangular shape to an ellipsoid [11] and a crease and two lobes appear. Another important contribution of the pericarp is to serve as a transient storage tissue. In addition, pericarp contributes to caryopsis feeding with the direct products of pericarp photosynthesis, and/or nutrients remobilized from disintegrated tissues. Its vascular tissues also allow for the supply of water and nutrients to the caryopsis. Moreover, it has been proposed that pericarp could set the upper limit of the caryopsis expansion and filling capacity by constraining the growth [12,13,14].

After fertilization, the caryopsis pericarp develops from the ovary wall and includes the epicarp or pericarp outer epidermis, the mesocarp and endocarp, or inner pericarp. The pericarp tissues undergo an important phase of cell elongation. From four days after anthesis (DAA), the mesocarp parenchyma cells undergo programmed cell death (PCD) [15]. The endocarp differentiates into a photosynthetically active chlorenchyma layer. The endocarp contains two cell layers known as cross cells and tube cells. At the end of the filling phase, the outer layers (pericarp, testa, and nucellar epidermis) are drastically reduced to only a few cell layers. The endocarp remains photosynthetically active until late development. While several studies were conducted to improve our knowledge on the development of the wheat pericarp [14,16,17], a comprehensive study with an assessment of the developmental and spatial variability of the pericarp anatomy in the entire caryopsis is lacking.

The development of plant tissues may be investigated through microscopy imaging. Optical microscopy on cross-sections reveals changes in the morphology at the cell scale, and may depict changes in the chemical composition [17,18,19]. Electron microscopy provides an enhanced resolution, but within a reduced field of view [20,21,22]. However, as the growth of organs is neither isotropic nor homogeneous, the changes in the morphology must be investigated in three dimensions and within the whole caryopsis. The three-dimensional anatomy can be investigated from the acquisition of many serial cross-sections, followed by a tedious reconstruction procedure [23,24]. An alternative is to use three-dimensional imaging, such as confocal microscopy [25,26,27]. However, the field of view and the small imaging depth limit this technique to small organs or tissue portions.

X-ray microcomputed tomography (µCT) is increasingly used for the non-destructive investigation of the 3D architecture of biological specimens, without requiring staining, sectioning, or inclusion [28]. Dry specimens present a high contrast to X-rays, but in the case of developing grain, tissues are highly hydrated and present a relatively low contrast to X-rays, making the tissues difficult to distinguish [11]. The coherent beam light provided by synchrotron facilities allows for phase contrast µCT and allows for the differentiation of some tissues within the organs [29,30,31]. The interest of synchrotron imaging for agronomy and food sciences research was recently highlighted in [32].

We have shown that X-ray µCT is a relevant technique for in vivo 3D imaging of the developing wheat caryopsis [11,31]. However, besides the proof-of-concept of the adequacy of the synchrotron micro-tomography for the live imaging of plant tissues, little was concluded about the provided information in relation to caryopsis development and in particular to the pericarp changes. The aim of this study was then to take advantage of synchrotron imaging to investigate changes in the morphology and anatomy of the pericarp, which has been hypothesized to be a central actor in caryopsis morphogenesis. For this, the first step was to validate the use of synchrotron micro-tomography to depict changes in the 3D morphology of the whole caryopsis. The second step was to identify tissues within synchrotron micro-tomography images that cannot be observed with laboratory tomography. The third step was to investigate the changes in the morphology at the cell level, focusing first on the epicarp, then on the mesocarp, and finally on the endocarp. The observations were then discussed in relation to the putative roles of pericarp in the caryopsis growth and development.

## 2. Results

### 2.1. Changes in Caryopsis Shape with Development

Phase contrast X-ray µCT images obtained with low and high resolutions were visualized using the ImageJ software and thoroughly examined. The images contained the caryopsis samples but also signals due to the tape used to fix the samples, and to the tube where the samples were inserted. These non-caryopsis items were removed, and the 3D volume of the caryopsis samples was reconstructed. Figure 1A–I shows a representative 3D volume reconstruction of wheat caryopses at the investigated stages of development, accompanied by examples of virtual cross and longitudinal sections. On virtual sections, an artefact due to the detector and known as the “ring artefact” was often observed in the center of the images, sometimes hampering the detailed observation of some tissues (Figure 1A).

The 3D representations highlighted the changes occurring in the caryopsis shape during its development with a major phase of elongation, an enlargement at the top, a deepening of the crease, an increase in the volume, and a collapse of the rolls at the top of the caryopsis. Virtual sections showed the growth of the seed within the developing wheat caryopsis, highlighted by the voids surrounding the seed. These voids were not detected at anthesis; they were visible first at the top of the caryopsis at 25 °DAA (Figure 1B) then spread to surround the seed at 100 °DAA (Figure 1E). Simultaneously, with the growth of the seed, the thickness of the pericarp (mainly mesocarp) tissue decreased (Figure 1).

### 2.2. Spatio-Temporal Variability of Tissues within the Whole Caryopsis

Low-resolution images of virtual cross-sections localized at different positions along the longitudinal axis of caryopses (Figure 2A) highlighted spatial variability in the tissue composition. For instance, at stage 50 °DAA at the bottom position, the caryopsis contained pericarp and vascular tissues; at the middle position, the space was mainly occupied by the embryo sac and a thick mesocarp, while the very top contained only pericarp tissues (Figure 2B). At stage 250 °DAA at the bottom position, the caryopsis contained pericarp, vessels, and the embryo; at the middle position, the space was mainly occupied by the endosperm surrounded by a thin pericarp, while the very top contained only pericarp tissues and voids (Figure 2C). 

High-resolution images of virtual cross-sections localized at different positions along the longitudinal axis of the caryopses (Figure 3A) revealed individual cells in several tissues (Figure 3B,C). In the pericarp, some cells of the mesocarp and endocarp (cross cells and tube cells) were distinguished due to intercellular spaces (Figure 3C). In contrast, the cells inside the embryo sac or later in the seed tissues were difficult to visually separate as there was no void between them (Figure 3B).

High-resolution images revealed other developmental variations. Our X-ray tomography acquisitions showed developing cross and tube cells from 50 °DAA highlighted by the formation of intercellular spaces. A gradient of differentiation was observed at 50 °DAA between the top of the caryopsis where cross and tube cells were distinguished, the middle region, and the bottom of the caryopses where they were hardly visible (Figure 3B). In virtual cross sections tube cells appeared as round cells while cross cells were elongated (Figure 3).

In addition, high-resolution images revealed spatial variations in the tissue and cellular morphology. For a given longitudinal position, variations were observed between the three regions imaged at a high resolution: the dorsal, lobe, and ventral regions (Figure 4A). At 250 °DAA, in the endocarp, tube cells were detected only in the dorsal position, as described in [20], and for all the observed longitudinal positions (Figure 4B). The mesocarp was almost totally disintegrated at the lobe position (Figure 4C) while it was still well represented in the ventral region (Figure 4D). 

The spatio-temporal variability observed in the pericarp prompted us to explore the potential of 3D reconstructions and image analysis in more detail to reveal new characteristics in the different pericarp tissues.

### 2.3. Cell Morphology within Epicarp

A careful observation of the surface of the caryopses obtained from µCT images using a 3D viewer revealed several anatomical details. For instance, at the top of the caryopsis, the basis of the stigmas and trichomes were visible. Trichomes covered almost half of the caryopsis at the earliest stages (Figure 1). In the ventral region of the caryopses, dark dots were noticed (Figure 5A). Using high-resolution images obtained for the ventral region, it was possible to zoom in towards these dots. The 3D view of the epidermis surface showed typical stomata structures with stomatal pores and guard cells (Figure 5B). In virtual sections across stomata, wide intercellular spaces that could serve as an air chamber were found (Figure 5C).

In all the investigated images, stomata were observed only in the ventral region and in the upper part of the crease region. The number of stomata present on each caryopsis was estimated by monitoring the pores at the surface of the caryopses. It ranged from 0 to 89, with an important variability between caryopses and across the development (Figure 5D). At anthesis, the number of stomata was low, with some caryopses having no detected stomata. In subsequent stages, there was an increase in the number of detected stomata, which seemed to reach a plateau around 100 °DAA. At 250 °DAA, the detection of stomata in the crease was difficult. This was due to the topology of the surface with an increased curvature and both lobes touching each other at the top and bottom of the caryopsis. The evolution of the number of stomata per caryopsis could be related to that of the whole caryopsis dimensions (Figure 5E), and, in particular, to a caryopsis length of up to 100 °DAA. Indeed, an increase in both the stomata number and length was observed between 0 and 50 °DAA, followed by a stagnation and then a sharp increase between 80 and 100 °DAA. 

In addition to stomata, our 3D images also revealed classical epidermal cells at the surface of the epicarp (Figure 6). These cells were visible only in the ventral region where the caryopsis samples were not adhering to the tape during the acquisitions (Figure 6A,C,F). Therefore, epidermal cells could not be comprehensively investigated in relation to their potential variability; in particular, no quantitative and kinetical study of the cell area could be provided. In young grains (50 °DAA), epidermal cell borders were hardly distinguished on 3D reconstructions of X-ray µCT images (Figure 6B) while they were visible at 250 °DAA (Figure 6E). Figure 6E shows an illustration of the variability in the cell shape in the ventral region of a representative caryopsis at 250 °DAA. Stomata, rows of elongated cells, but also cells with a more globular shape were observed. This diversity of shape was confirmed by the examination of microscopy images of wheat caryopses at the same stage (Figure 6F,G). Microscopy images showed that epicarp cells have a more homogeneous and square shape at 50 °DAA (Figure 6D).

### 2.4. Porosity and Cell Orientation of Mesocarp

The mesocarp tissue contains parenchyma cells. As shown in Figure 1, Figure 2, Figure 3 and Figure 4, the mesocarp tissue contained large voids surrounding the seed due to programmed cell death, and smaller voids between mesocarp cells. X-ray image observation suggested a variability in the proportion of intercellular voids depending on the position within the caryopsis (Figure 1). In particular, we observed intercellular voids at the top of the grain (Figure 1) and in the ventral side of the caryopses below the epicarp (Figure 4 and Figure 5). This region was investigated using 3D visualizations (Figure 7A). This highlighted the connections between these empty spaces between ventral mesocarp cells (Figure 7B).

To quantify the intercellular spaces in the mesocarp and assess their variability, we implemented a procedure to calculate the porosity of the mesocarp tissue. Figure 8A depicts the regions used to compute the porosity. Porosity values (volume of voids/total volume) for individual images ranged from 0.0 to 0.7. Average porosity maps were constructed for caryopses at different stages and are presented in Figure 8B–E. They revealed that the ventral mesocarp porosity varies spatially. For a given development stage, it was higher in the upper central part of the caryopsis than at the bottom of the caryopsis. This reflects the presence of stomata in the upper central part of the caryopsis (Figure 5). Figure 8B–E also highlights developmental variations, with an increase in the porosity from 50 to 200 °DAA.

The observation of transverse virtual sections (Figure 9A) revealed an apparent heterogeneity in the mesocarp cell shape depending on the position in the caryopsis (Figure 9B–I). In caryopses undergoing elongation (100 °DAA), mesocarp cells at the middle part of the caryopses appeared to be round, while they were elongated at the top of the caryopsis (Figure 9E compared to Figure 9C). In longitudinal virtual sections, mesocarp cells appeared elongated in the middle of the caryopsis and slightly elongated at the top of the caryopsis (Figure 9H,I), and their elongation axis assigned to their biggest dimension was different. The elongation axis of mesocarp cells in the middle region was in line with the caryopsis elongation axis. At the top of the caryopsis, the elongation axis seemed in line with radial growth. 

The variability in the mesocarp cell shape/orientation between the top and middle regions was observed at all investigated stages from anthesis to 150 °DAA. At anthesis, mesocarp cells at the top were round to slightly elongated (Figure 9B). Cells were slightly elongated in the middle of the caryopsis, but a low intercellular space made them difficult to visualize (Figure 9D). At the bottom, cells were small and round (Figure 9F,G). Relatively more mesocarp cells were elongated in the direction of caryopsis elongation at 100 °DAA compared to anthesis. In subsequent stages, the mesocarp was too degraded to compare the cell shape and orientations. 

### 2.5. Cell Morphology and Organisation of Endocarp

Virtual sections highlighted the spatial and temporal variability in the cell shape of the endocarp (Figure 3 and Figure 4). To more globally explore the structure of the endocarp, its surface was isolated using the Crop3D software for caryopses at two stages of development: at 150 °DAA and at 250 °DAA in the lobe region. Before 150 °DAA, the mesocarp was not disintegrated enough to allow for a segmentation of the endocarp. 

At 250 °DAA, 3D reconstructions were performed at different positions along the caryopsis (Figure 10A). They highlighted the surface of the endocarp cross cell layer (Figure 10B–G). The surface was not flat, and the cross cells appeared as patches of continuous and elongated cells arranged in parallel with the ends located at approximately similar positions and forming ridges.

In the dorsal region of the caryopsis, below the cross cells layer, tube cells were present. Their 3D structure and arrangement were obtained using the Crop3D software at different longitudinal positions (Figure 11A). In stages prior to 150 °DAA, the tube cells were not clearly distinguished in the X-ray images in positions other than the upper part of the caryopsis because no sufficient voids were present in the cell layer (Figure 3B). The 3D representations highlighted the fact that tube cells were elongated and orientated in the same direction. Their orientation axis was perpendicular to that of the cross cells, both cell layers forming a 3D grid (Figure 11E–H). At 150 °DAA, tube cells were contiguous, although spaces were present between the cells at the top (Figure 11B,E). At the bottom and at the top (Figure 11B,D), the tube cells were mostly adherent to the seed, while in the middle part, the tube cells did not systematically adhere to the seed (Figure 11C). 

At the extremities (top and bottom) of the caryopses (Figure 12A), virtual slices revealed differences in the organization of the endocarp compared to the middle region. At the top, several layers of endocarp were visible and their elongation axis seemed orthogonal (Figure 12B). The 3D reconstruction of individual cells was carried out using the Free-D software to visualize the arrangement of these top endocarp cells compared to the underneath seed compartment (Figure 12C,D). At the bottom of the caryopsis, several layers of endocarp and structures forming “extensions” from the seed were also visible (Figure 12E). The 3D reconstruction of individual cells showed that the cells that appeared as “extensions” were cells with an elongation axis that followed the curvature of the seed, while cells in the next layers had an elongation axis corresponding to the caryopsis longitudinal axis (yellow and brown) (Figure 12F).

## 3. Discussion

The cereal grain or caryopsis contains several tissues with different functions, such as support, protection, storage, water, and nutrient transport. Several studies have been conducted to characterize the tissue structure of wheat caryopses; however, most studies examined only part of it in the cross-sections of caryopses observed by optical microscopy, fluorescence microscopy, and electron microscopy; e.g., as in [17,33]. Using synchrotron-based X-ray µCT, image analysis and 3D reconstruction we were able to explore entire caryopses in 3D at different stages of development at a low or high resolution. Our study revealed spatial (depending on the position in the caryopsis) and temporal (developmental) variability in caryopsis tissue organization, cell shape, and orientation, especially in the pericarp. X-ray µCT also has the advantage of not requiring fixation, dehydration, embedding, and cutting steps which damage the tissues. However, it has drawbacks since it does not allow us to distinguish all the tissues and cells, nor to have access to the inside of cells even with a high resolution and high signal/background ratio obtained in the synchrotron facility [31]. Moreover, our experimental settings generated additional difficulties and missing data. The use of fixing tape with a high X-ray signal hampered the comprehensive survey of the caryopsis surface and the selected regions for high-resolution µCT did not allow us to explore fully the very top and bottom of the caryopses.

However, new observations for the different tissues of the pericarp were obtained and their relations with the processes governing caryopsis growth are discussed below.

### 3.1. New 3D Anatomical Features Revealed in Epicarp and Mesocarp

At the caryopsis surface, we were able to observe epicarp cells and distinguish different types of epidermal cells. Normal “pavement” cells were detected with a rectangular shape and elongation axis following that of the elongation axis of the caryopsis. We could not explore the spatial variability in epicarp pavement cells due to our experimental settings. Variability has already been reported in the shape and dimension of wheat epicarp cells. They were described as elongated cells following the longitudinal axis, with narrower cells in the crease region and less elongated cells (almost isodiametric cells) at the bottom part of the caryopsis [34]. Cells with a more globular shape were also detected. They were also visible in the cross-sections observed by microscopy. Plant epidermal cells can arbor many shapes. For instance, the epidermis of the stamens of *Clematis macropetala* comprises nine epidermal cell types differing by their micromorphology [35]. The wheat epicarp globular cells might have a specific function. We also observed specialized cells. Trichome cells are easily visible on the ovary and on top of the caryopsis, as described in the literature [36]. The guard cells of stomata were detected only in the upper ventral region of the caryopses. Stomata had already been reported in cereal grains [21,34,37], but in the recent literature, they are hardly mentioned.

Underneath the epicarp, mesocarp cells are parenchyma cells revealed as elongated cells with axes of elongation different at the top and at the middle part of the caryopsis. From 0 to 100 °DAA, the number of cells elongated in the same direction as the caryopsis length growth increases. This may reflect a probable role of mesocarp cells in caryopsis elongation at these stages. Cells located above the rolls and at the bottom of the caryopsis are not likely to be involved in caryopsis elongation/length since they are not elongated longitudinally. However, mesocarp cells at the top of the caryopsis were found to be elongated in different directions, such as towards the lobes. They might be involved in the expansion of the lobes and the formation of the caryopsis shape. Then, in the mesocarp tissue, large voids were detected; these voids result from mesocarp PCD, an important process in cereal caryopsis development which takes place to make room for the developing seed [15]. Voids were detected as soon as 25 °DAA (around 1 DAA), earlier than has been described in the literature. Moreover, our study showed that PCD started at the top/apical end of the caryopsis. The mesocarp cells in the dorsal and lobe regions were almost all degraded at 250 °DAA, while they remained intact in the ventral region as already described in [17]. In addition, X-rays highlighted intercellular spaces in the mesocarp tissues. Variability in the level of air space was found between developmental stages and between different positions within the caryopsis. We uncovered mesocarp with a high intercellular porosity in the ventral crease region where stomata were detected. Within the aerated mesocarp, vascular tissues were detected by X-rays as a dense area with no intercellular spaces. The ovary vascular bundles present in both lobes remain visible in tomography images up to the end of early development. Contrary to mesocarp cells, vascular cells are not degraded. The caryopsis vascular tissues are also visible as a dense area, but few individual cells are revealed by X-rays inside the tissue.

The endocarp derives from the cell layers of the ovary that surround the ovule. The outer layer differentiates into cross cells. Three-dimensional reconstructions allowed us to visualize the endocarp in 3D and showed that the cross cells surround and envelop the seed and form patches of elongated cells. They have a more elongated shape in the ventral region than in the dorsal region [17] and, as we see in this work, in lobe regions. The inner layer differentiates into tube cells which are only present in the dorsal region [20]. This restricted localization is sometimes omitted in recent publications. Our observations showed that this layer exhibits a gradient of differentiation from the top to the bottom of the caryopsis; at 50 °DAA, the cells are well distinguished only at the top of the caryopsis. In 3D, we showed a variability in tube cells arrangement between the top, middle, and bottom of the caryopsis, with cells being more adjacent at the bottom of the caryopsis, and less adherent to the seed in the middle of the caryopsis. Tube cells and cross cells have an orthogonal elongation direction [17,20] and where both cell layers are present, they form a 3D grid. At both caryopsis extremities, several layers of endocarp cells were detected. They also have different elongation axes. We gathered new data to study the 3D arrangement of pericarp cells; however, our experimental design did not allow us to survey the entire diversity in cell arrangement at the very top and bottom of the caryopses.

All the gathered information was examined in relation to the role of pericarp in the grain growth.

### 3.2. Caryopsis Elongation, Radial Expansion, and Physical Constraints

#### 3.2.1. Mesocarp Cells and Growth

Our study highlighted the fact that in early development, mesocarp cells have different elongation directions and that the number of longitudinally elongated cells increases underneath the rolls between anthesis and 100 °DAA. A schematic model of the main elongation axis of mesocarp cells related to the caryopsis growth is proposed in Figure 13. Above the rolls, mesocarp cells have an elongation axis in different directions depending on their position. Then, most mesocarp cells undergo PCD. This highlights a crucial role of mesocarp cell elongation above the rolls for the shape of the top of the ovary/caryopsis. Similarly, below the rolls, a crucial role of mesocarp cell elongation is suggested for caryopsis elongation (and caryopsis length) at early stages only. Such an important role of the mesocarp was reported in [38] who compared fertilized and unfertilized sterile wheat ovaries in terms of their dimensions, shape, and anatomy. They showed that in contrast to fertilized ovaries, which essentially grow vertically, unfertilized ovaries grow only radially. This resulted in ovaries with tops larger than that of fertilized caryopses, rounder “grains” with very shallow crease and less marked lobes (Figure 2E in [38]). In these sterile ovaries, no PCD was detected in mesocarp cells; the authors propose that fertilization induces mesocarp PCD which regulate the mesocarp growth.

#### 3.2.2. Putative Role of the Endocarp

The endocarp forms a tissue adhering more or less tightly to the seed. It covers the seed inside the developing caryopsis, while at certain stages due to mesocarp PCD, it is not in contact with the rest of the pericarp: mesocarp and epicarp. The layer of cross cells is doubled with a layer of tube cells in the dorsal region, forming a grid. The endocarp forms a grid also at the seed’s extremities. Such grids might act as a physical support or reinforcement. They might also locally constrain/control the growth of the seed within the caryopsis (Figure 14). They might be constraining the growth longitudinally at both extremities and radially in the dorsal part of the caryopsis. Interestingly, the rice caryopsis, which has a different shape than wheat caryopsis, contains tube cells surrounding the seed and not only in the dorsal area [39].

The voids separating the outermost layers of the caryopsis and the seed+endocarp question the relationship between the expansion of the caryopsis outer layers and that of the seed. The importance of outer epidermal cell layers in determining organ size and shape is recognized as “epidermal-growth-control” [40,41]. Here, the seed growth does not seem to be systematically physically limited by the growth of the caryopsis epicarp due to physical separation. However, epicarp could set the size potential of the caryopsis. Depending on the stages of development, the endocarp may also contribute to the growth control of the caryopsis. The radial growth of the caryopsis in early development is not uniform, resulting in the formation of two lobes and of a deep crease. The genesis of this complex shape requires physical constraints. In the crease, these constraints may be determined by the presence of the dense zone of the vascular bundle. In the dorsal region, they may be determined in part by the grid of the endocarp (Figure 14). The less constrained regions in between could expand more freely under the effect of the internal pressure of the developing seed and give rise to the lobes. Additional studies targeting the *Triticum* species with a different caryopsis size and shape, as described in [42], would be of interest to address the contribution of the pericarp tissues. Combined with modelling these studies would help to better understand the formation of the complex shape of the wheat caryopsis. 

#### 3.2.3. Putative Role of the Pericarp in Caryopsis Photosynthesis

The presence of stomata in the ventral crease area, close to the vascular bundle of the wheat caryopsis, has already been reported [20,21,34,37] in a density close to that reported here [21]. However, as a new finding in this study, we showed that there is an increase in the stomata number at the early stages of development, particularly between 0 and 100 °DAA when the caryopsis growth is particularly rapid and significant. Moreover, our study reports large airspace and space connections in the chloroplast-containing mesocarp behind wheat caryopsis stomata; the volume of airspaces, estimated by porosity values, increases between 0 °DAA and 200 °DAA. These mesocarp porosity values are similar to maximal mesophyll porosity values found in leaves (0.25 to 0.35 depending on wheat lines) [43]. Note that our values could have been underestimated due to our calculation method, which includes the epicarp cell layer. Mesophyll airspace formation has been proposed as a determining factor to the stomatal function in both monocots and eudicots [43]. 

Our experimental set-up did not allow us to assess the functioning of the stomata nor any direct measurement of photosynthesis. However, provided that the stomata are functional, our results support the hypothesis of the photosynthetic capacity of the pericarp and its ability to contribute to the assimilate transfer to the developing grain. Pericarp photosynthesis would then provide the energy and oxygen required for the early growth of caryopsis. Note that our results do not prejudge the origin of CO_2_: external CO_2_ or a possible recycling of endogenously CO_2_ released by grain respiration [44]. This hypothesis does not exclude completely the possibility of gas exchanges occurring via the thin cell walls of wheat caryopsis outer tissues [17].

Given the presence of functional chloroplasts and the transient storage of assimilates (starch, [17]), the presence of stomata and aerated parenchyma in the vicinity of vascular tissues, and a possible combination of C3 and C4 photosynthesis in the wheat pericarp [45], it seems important to re-investigate the contribution of pericarp photosynthesis to the growth of cereal caryopsis. 

### 3.3. Towards a 3D Model of Caryopsis Morphogenesis

The images obtained with the two experimental setups (whole caryopsis imaging and high-resolution imaging) provide complementary information. When imaging the whole caryopsis, it is possible to depict the global shape of the caryopsis, and the morphology (thickness and porosity) of some tissues can be described. The morphology of cells is, however, still difficult to assess. Using high-resolution imaging allows us to investigate the 3D morphology of thin tissue layers (e.g., tube cells and cross cells), and the morphology of a larger number of cells in more detail. However, due to the size of the field of view, the observations and measurements are more difficult to generalize to the whole caryopsis. Another point worth mentioning is that 3D micro-tomography provides information solely on the morphology of cells and tissues. Other imaging techniques can provide complementary information. For example, magnetic resonance imaging allows for quantifying water fluxes within the organ. Microscopy or mass-spectroscopy imaging can describe the chemical composition within tissue slices. Moreover, the local mechanical properties of tissues can be measured using atomic-force microscopy. 

The joint analysis of the information provided by different techniques represents a methodological challenge. Correlative imaging approaches and image registration methods represent a first technical solution to combine different observations from the same sample [46,47,48]. An additional difficulty comes from the fact that imaging data are often acquired on different samples. In that case, the information provided by an individual image has to be interpreted as a sample measurement representative from a whole population.

A promising way for the capitalization and the fusion of the information provided by various imaging devices or setup is to consider the construction of synthetic atlases of biological organs. This was successfully applied in medical sciences for human organs such as the brain, heart, or liver [49,50,51]. Based on micro-tomography data of wheat caryopsis, the first step could be the constitution of a 3D atlas of the whole caryopsis images by applying a group-wise shape registration strategy at each development stage. In the second step, the integration of high-resolution images could enrich the atlas with cell morphology information. The constitution of this 3D+time morphological atlas will later allow for registering images from other imaging techniques. This will allow for a consideration of the evolution of the physical and chemical composition of tissues within the caryopsis and pave the way for an integrated vision of plant organ development.

In conclusion, using X-ray µCT, in vivo 3D imaging highlights the important spatial and temporal variability of pericarp tissues and cell morphology of the growing wheat caryopsis. First, from anthesis to 150 °DAA, we showed an increasing number of stomata in the epicarp of the upper ventral region of the caryopsis, next to the crease; it is accompanied by an increasing porosity of mesocarp tissue underneath the stomata. This supports the idea of a major role of the pericarp in caryopsis photosynthesis. Second, 3D reconstructions of the growing caryopsis showed a different mesocarp cell orientation between the top and the middle of the caryopsis. This difference increases from anthesis to 150 °DAA. Moreover, 3D images highlight the presence of a grid formed of cross and tube cells only on the dorsal region, whereas cross cells cover the whole surface of the seed. All together, these observations may contribute to a better understanding of the anisotropic growth of the wheat caryopsis and of the physical constraints involved in its final shape and size. Finally, these results should be of interest both for the researchers interested in caryopsis development and physiology, and more generally for cereal science and the whole plant development community.

## 4. Material and Methods

### 4.1. Plant Material and Preparation of Samples

The wheat (*Triticum aestivum* L.) plants cultivar Recital were cultivated from sowing to a few days around anthesis in a controlled greenhouse at INRAE Clermont-Ferrand, where the mean day temperature was 16.0 ± 1.8 °C and under natural daylight, supplemented to a 16h photoperiod. Plants were watered daily in excess twice a day. Around anthesis, plants were transported to the synchrotron SOLEIL for X-ray tomography acquisitions which were conducted as described in [31]. There, plants were kept at 15.2 ± 1.4 °C for the 4 days of image acquisitions. For each spike, the date at which anthers become visible (anthesis) for the florets of the middle spikelets was noted to estimate the development by calculating the cumulated average day temperature (with 0 °C as the base temperature) since anthesis in degrees days after anthesis: °DAA). The basal caryopsis of the two middle spikelets were harvested at 9 developmental stages immediately before the acquisitions. The samples covered the early phase of caryopsis development from 0 to 250 °DAA, i.e., anthesis to the beginning of the filling phase. Number of replicates are listed in Table 1.

Representative caryopsis samples at the targeted stages were imaged using a stereomicroscope (SMZ800 Nikon), and some samples were prepared as described in [52]. In short, they were fixed in 1% (v/v) glutaraldehyde and 3% formaldehyde in 0.1 M of phosphate buffer, pH 7.4, then dehydrated through a graded aqueous ethanol series (30, 50, 70, 85, 95, and 100%). The samples were then embedded in London Resin White. Semi-thin sections were prepared (1 µm, ultracut UC7, Leica), stained with 0.1% toluidine blue, and observed using a multizoom macroscope under bright-field conditions (AZ100 M Nikon) [53]. 

### 4.2. Image Processing and Analysis

#### 4.2.1. Image Preprocessing

Image data generated by the Psiche beamline consisted in binary data files in a raw format, with intensity coded into 32-bits floating point values. Raw data files were converted to 3D 8-bits grayscale images, with the intensity re-calibrated between 0 and 255, and saved as 3D Tiff images. For each 3D image, three 2D slices corresponding to the middle slice along each of the X, Y, and Z direction were generated to provide a quick overview of the caryopsis shape. Preprocessing was performed with the ImageJ/Fiji software [54].

#### 4.2.2. Caryopsis Segmentation

The 3D images acquired at low-resolution contained the caryopses, but also the materials used to fix the caryopses during acquisition (tube, plastic tape, rubber tape). The 3D binary images of the caryopses were segmented using the workflow described in [31]. In short, the bright structures were identified using a manual threshold of the volume. Morphological filtering operations (opening, closing, holes filling, and morphological reconstructions) were applied to separate the caryopsis from the tube, and to remove the voxels belonging to the structures that maintained the caryopsis. Image processing resulted in 3D binary images where the foreground corresponded to the wheat caryopsis region. Segmented images were used as binary masks to generate 3D grayscale images retaining only the voxels belonging to the caryopses and replacing background voxels with the value 0.

#### 4.2.3. Alignment

To facilitate further analysis, each caryopsis was aligned along its main axes. First, the crease tip was identified from segmented cross-sections of the caryopsis. The series of 3D positions were used to align the crease axis with the vertical axis. In the second step, the caryopsis was rotated around the vertical axis such that the two lobes were orientated in the y-axis direction. Finally, each caryopsis was centered within the image, using the same image size for all the caryopses of the same stage. The same alignment procedure was applied to binary images and grayscale images. The estimation of the transform parameters was performed within Matlab (the Mathworks, Natick, MA, USA). The 3D aligned images were generated using in-house software tailored to process large 3D images (https://github.com/SciCompJ/Imago, accessed on 21 February 2023).

#### 4.2.4. Visualization

The ImageJ/Fiji software [54] was used for the interactive visualization of 3D images, either using slice-by-slice display, or using the 3D Viewer plugin. To facilitate the interpretation of 3D structures, a specific plugin for ImageJ was designed that allowed for the generation of a 3D scale bar with various locations and orientations (https://github.com/ijtools/ScaleBar3d, accessed on 21 February 2023).

#### 4.2.5. Morphometry

The global morphometry of the caryopses was measured from 3D binary images, using the same strategy as in [11]. To measure the global shape of the caryopsis, morphological filtering operations were applied to remove the hairs of the caryopses and to fill the voids within the caryopsis. The dimensions of the caryopses were measured by computing the difference between the minimum and maximum extent in each dimension after alignment.

#### 4.2.6. Pericarp Porosity

The pericarp porosity was quantified on grayscale images after 3D alignment. A representative portion of the XY slices was selected, and the contour of the caryopsis was retrieved from the binary slices and smoothed. The portion of the contour comprised between the two lobe extremities was retained and divided into two halves. A fixed number of ten regions perpendicular to each curve was determined by dividing each curve into equal-length portions and considering the parallel to each curve. The parallel curve was obtained by applying a translation in the direction perpendicular to the curve and with a distance chosen according to the average depth corresponding to each stage. The porosity was measured within each region of each slice by counting the proportion of pixels within the grayscale slice with a value below the given threshold. A threshold value equal to 140 was chosen. The values on all the slides were summarized into a table, with 50 rows corresponding to the relative heights within the grain, and 20 columns corresponding to the relative horizontal position on the crease. The whole workflow was implemented within Matlab (the Mathworks, Natick, MA, USA) and is available upon reasonable request.

#### 4.2.7. Three-D Visualization of the Organization of the Inner Tissues

Several tissues and cell layers within the caryopsis structure are located inside or between other bright tissues, making it difficult to assess their 3D structure and organization. For this, we developed a semi-interactive software tool (Crop3D) that allows us to crop from a 3D image a region corresponding only to the desired tissues. The user is able to select representative slices and manually extract the 2D contour corresponding to the structure of interest. The contours are merged together with help of Fuchs’ algorithm [55] to generate a 3D polygonal mesh that surrounds the structure. Finally, the result image is generated by using the intensity of the original image for voxels located inside the mesh, and zero for voxels located outside. The development was integrated into the Imago software, which is freely available at https://github.com/SciCompJ/Imago (accessed on 21 February 2023).

At the cell level, the Free-D software [56] was used for the manual reconstruction and visualization of the 3D cell within 3D images.

#### 4.2.8. Visualization of the 3D Epicarp (RotCrop3D)

The 3D visualization of epicarp cells from high-resolution 3D images is made difficult by the variations in the orientation of the epicarp, the size of the images, and the potential presence of other structures (tape and surrounding tissues). To overcome these problems, we developed the RotCrop3D plugin for the ImageJ/Fiji software. RotCrop3D allows for cropping and resampling a portion of a 3D image using arbitrary orientation. The user can select the size of the result image, the center of the box, and the orientation. The orientation can be chosen from three rotation angles. It can also be evaluated automatically by computing the local gradient around the center point and using its direction for the z-axis. The result is a new 3D image, where the epicarp is roughly parallel to the XY plane, making the visualization of epicarp cells and stomata much more straightforward. The plugin is freely available from https://github.com/ijtools/ijRotatedCrop (accessed on 21 February 2023). 

## Figures and Tables

**Figure 1 plants-12-01038-f001:**
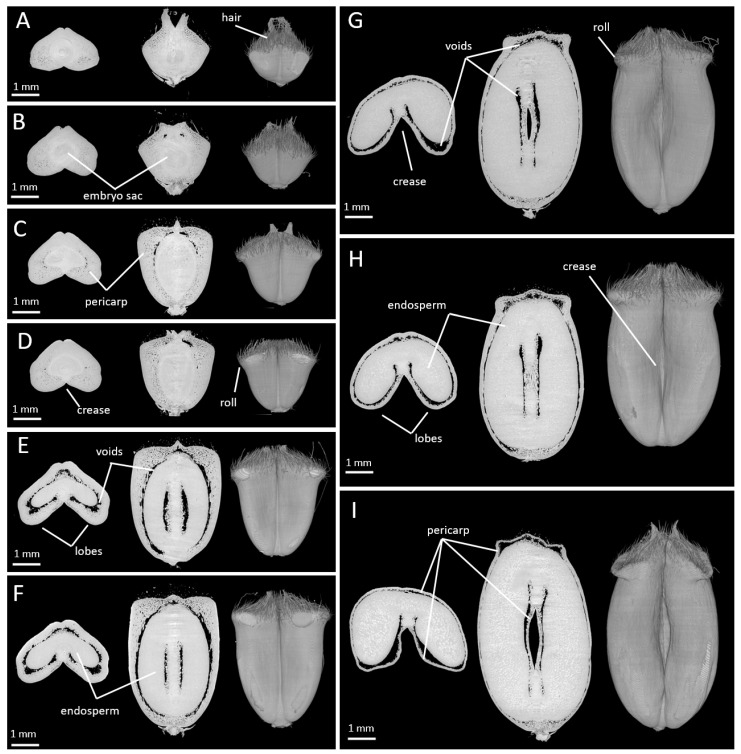
Representative X-ray tomography images obtained with low resolution for wheat caryopses at stage 0 to 250 °DAA. 0 °DAA (**A**), 25 °DAA (**B**), 50 °DAA (**C**), 80 °DAA (**D**), 100 °DAA (**E**), 150 °DAA (**F**), 180 °DAA (**G**), 200 °DAA (**H**), 250 °DAA (**I**). For each stage, from left to right, examples of virtual cross and longitudinal sections sampled from the 3D caryopsis reconstruction.

**Figure 2 plants-12-01038-f002:**
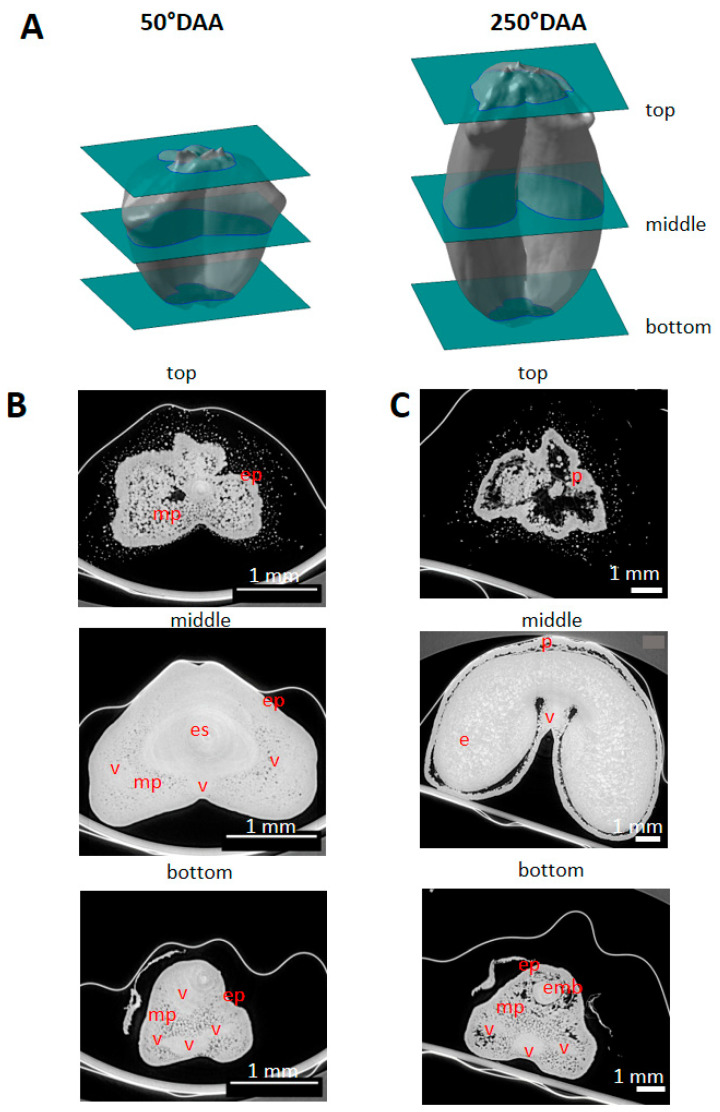
Spatial variability in tissue composition observed along the caryopsis longitudinal axis. (**A**) Localization of the virtual sections (green) from low-resolution X-ray µCT. (**B**) Representative image in three regions of a wheat caryopsis harvested at 50 °DAA. (**C**) Representative image in three regions of a wheat caryopsis harvested at 250 °DAA. Legends: mp—mesocarp; ep—epicarp; v—vascular tissue; e—endosperm; emb—embryo; es—embryo sac.

**Figure 3 plants-12-01038-f003:**
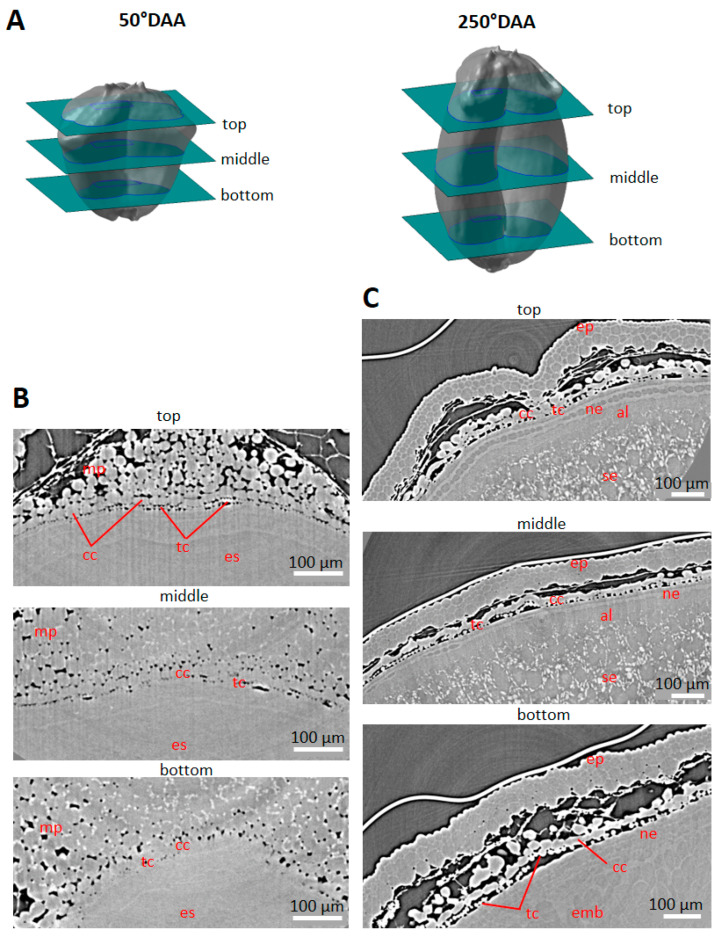
Spatial and developmental variability observed along the caryopsis longitudinal axis. (**A**) Localization of the virtual sections (green) and shown regions (blue rectangles) from high-resolution X-ray µCT. (**B**) Representative image in three regions of a wheat grain harvested at 50 °DAA (dorsal area). (**C**) Representative image in three regions of a wheat grain harvested at 250 °DAA (dorsal area). Legends: mp—mesocarp; cc—cross cells; tc—tube cells; ep—epicarp; se—endosperm; emb—embryo; ne—nucellar epidermis; al—aleurone; es—embryo sac.

**Figure 4 plants-12-01038-f004:**
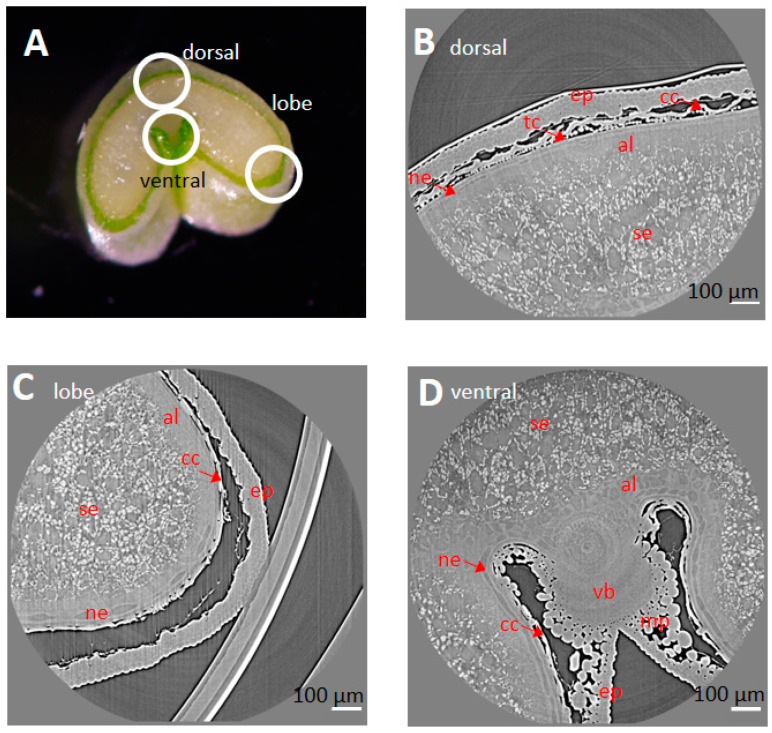
Spatial variability observed in different regions on a wheat section at the equatorial position. (**A**) Regions targeted for high-resolution X-ray µCT on a transverse section. (**B**–**D**) Representative tomography image in dorsal (**B**), lobe (**C**), and ventral (**D**) regions of a wheat caryopsis harvested at 250 °DAA. Legend: mp—mesocarp; cc—cross cells; tc—tube cells; ep—epicarp; vb—vascular bundle; se—starchy endosperm; ne—nucellar epidermis; al—aleurone.

**Figure 5 plants-12-01038-f005:**
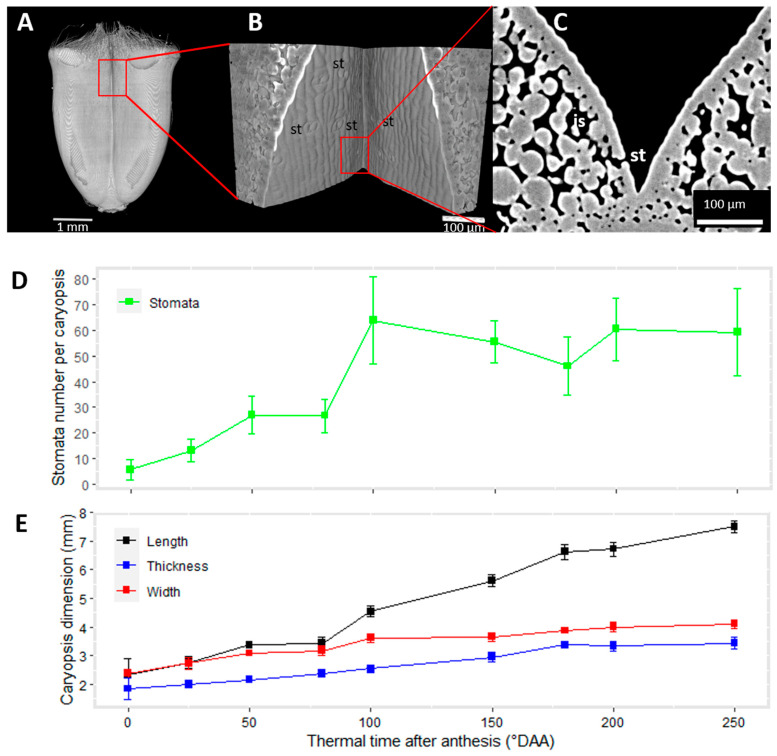
Stomata are present on the ventral face of wheat caryopses. A 2D projection of a 3D view of the caryopsis surface showing stomata in the ventral region of a caryopsis at 100 °DAA (**A**). Zoom on the epicarp of the caryopsis in the region depicted by the red frame (**B**). Virtual section across a stomata showing the guard cells and the substomatal chamber (**C**). Concomitant evolution along time of stomata number per caryopsis (**D**) and the whole caryopsis dimensions (**E**). Legend: st—stomata; is—intercellular space.

**Figure 6 plants-12-01038-f006:**
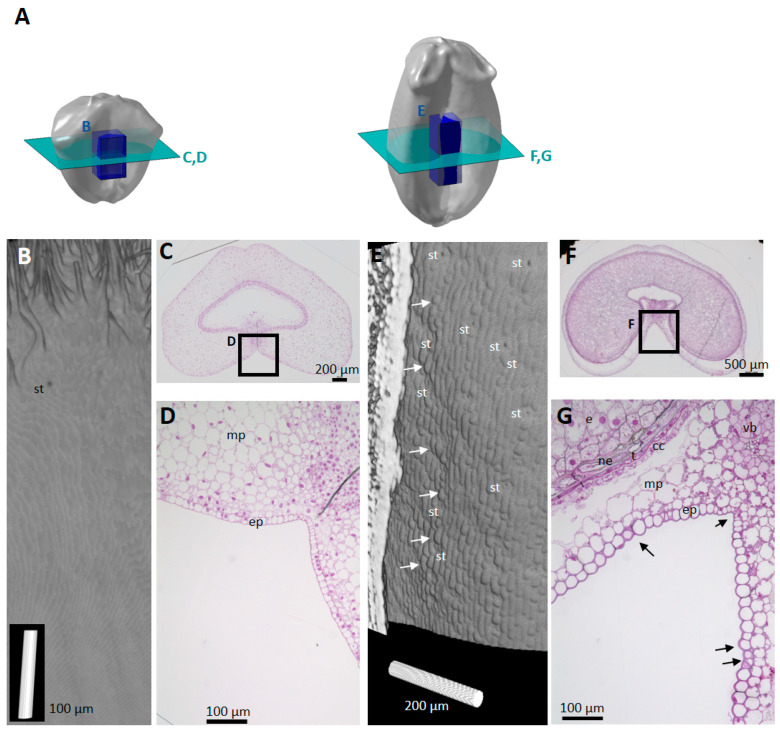
Variability in epicarp cell shape. (**A**) Localization of the regions observed. Light blue planes correspond to approximate positions of microscopy sections shown in (**C**,**D**,**F**,**G**). Dark blue boxes correspond to the position of 3D reconstructions. (**B**,**E**) A 2D projection of a 3D visualization of epicarp cell surface from X-ray µCT, ventral side, 50 °DAA (**B**) 250 °DAA (**E**). Cell borders are visible at 250 °DAA not at 50 °DAA. (**C**,**D**) Light micrographs of a wheat caryopsis harvested at 50 °DAA. (**C**) Entire cross-section stained with toluidine blue, and (**D**) zoom on the epicarp cells in the ventral region. (**F**,**G**) Light micrographs of a wheat caryopsis harvested at 250 °DAA. (**F**) Entire cross-section stained with toluidine blue, and (**G**) zoom on the epicarp cells in the ventral region. The arrows point at different cell shape and size. Legend: mp—mesocarp; cc—cross cells; tc—tube cells; ep—epicarp; vb—vascular bundle; e—endosperm; ne—nucellar epidermis; st—stomata; t—testa.

**Figure 7 plants-12-01038-f007:**
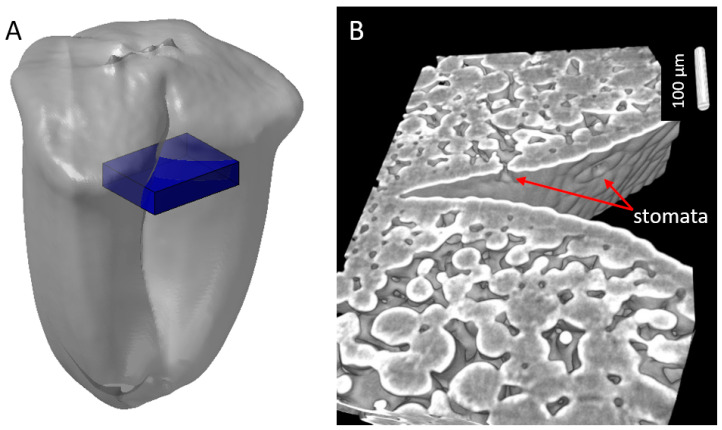
Representative example of the inter connections of empty spaces between cells within ventral mesocarp of a wheat caryopsis at 100 °DAA. (**A**) Localization of the region observed. (**B**) A 2D projection of a 3D view.

**Figure 8 plants-12-01038-f008:**
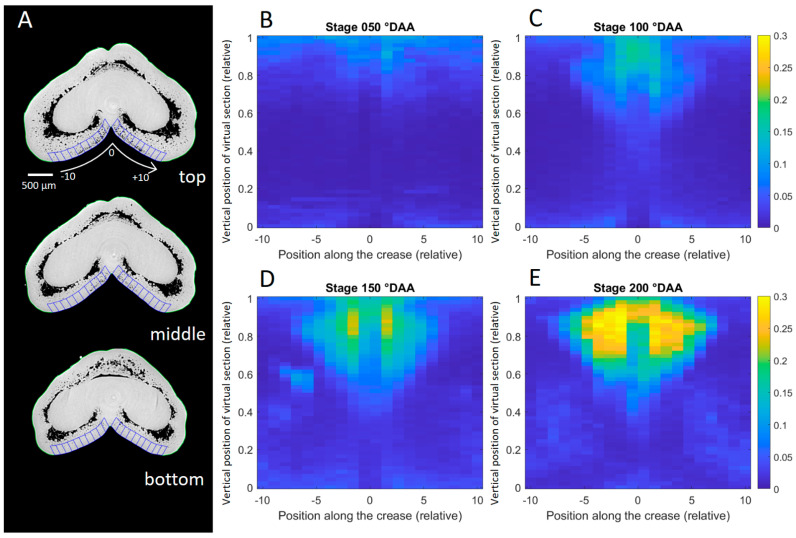
Variability in mesocarp porosity in the ventral region of the caryopsis. (**A**) Representation of the measurement regions for three virtual sections within a caryopsis at stage 150 °DAA. (**B**–**E**) Average porosity maps obtained from all the caryopses at 50, 100, 150, and 200 °DAA (n = 5 to 10). Porosity was calculated as ratio of volume of voids/total volume of targeted regions delineated in blue in A. The regions were located such that they divide the ventral face into 20 regions on each side around the crease, from −10 to +10. This calculation was repeated along the longitudinal axis and normalized between 0 (bottom of the caryopsis) and 1 (top of the caryopsis). The colors correspond to the porosity values, between 0 (no void, blue) and 0.3 (yellow).

**Figure 9 plants-12-01038-f009:**
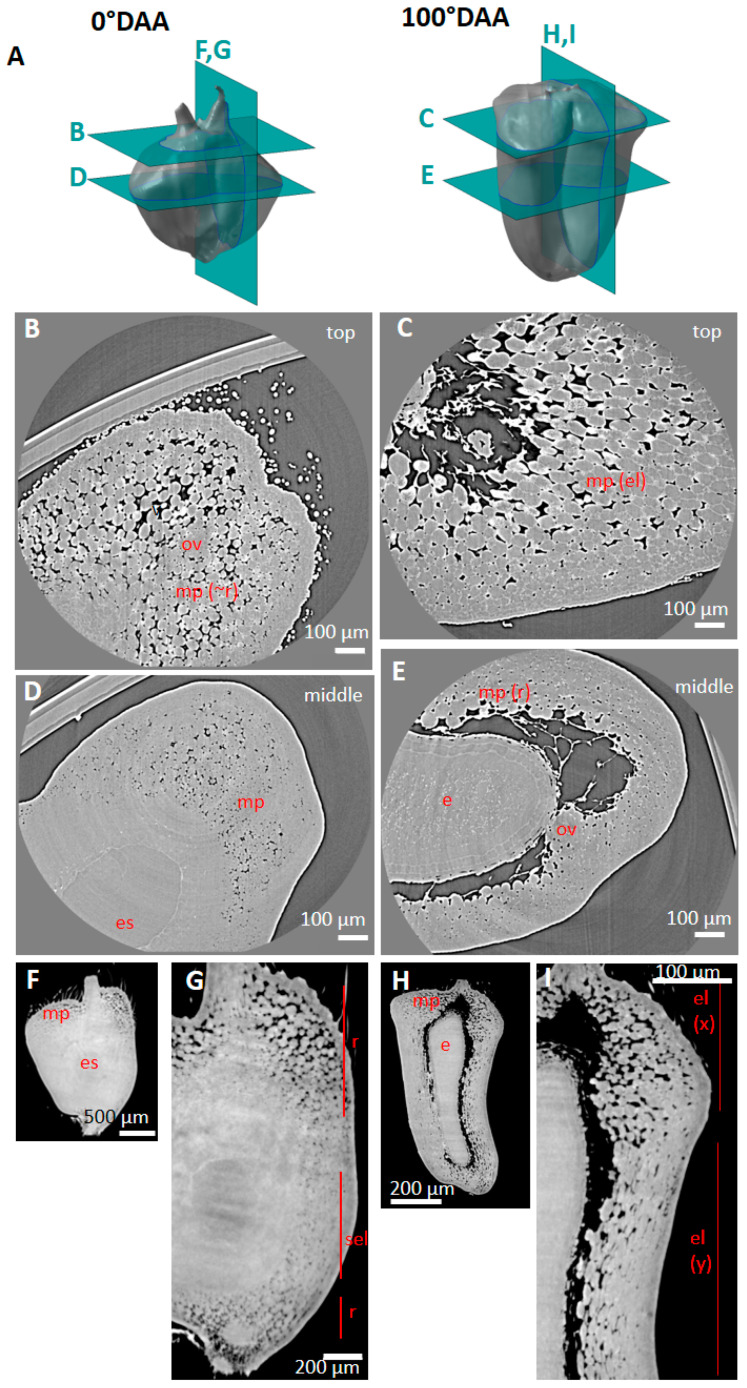
Spatial and developmental variability in mesocarp cell orientation observed along the caryopsis longitudinal axis. (**A**) localization of virtual sections. (**B**,**C**) Representative high-resolution tomography image (transverse virtual section) at the top of a wheat caryopsis harvested at anthesis 0 °DAA (**B**) and 100 °DAA (**C**) (lobe area). Mesocarp cells have an irregular shape, nearly round (r) at anthesis, elongated (el) at 100 °DAA. (**D**,**E**) Representative high-resolution tomography image (transverse virtual section) at the middle of a wheat caryopsis harvested at 0 °DAA (**D**) and 100 °DAA (**E**) (lobe area). Mesocarp cells have a round shape. (**F**,**G**) Representative low-resolution tomography image (longitudinal virtual section) of a wheat caryopsis harvested at 0 °DAA. (**G**) Zoom in (**F**) to highlight cell orientation. Mesocarp cells have a round (r) shape at the bottom and top, and a slightly elongated (sel) shape at the middle of the caryopsis. (**H**,**I**) Representative low-resolution tomography image (longitudinal virtual section) of a wheat caryopsis harvested at 100 °DAA. (**I**) Zoom in (**H**) to highlight cell orientation. Mesocarp cells have elongated shape at the top and middle of the caryopsis, but their elongation axis is in different orientations (x at the top and y at the middle). Legend: mp—mesocarp; cc—cross cells; e—endosperm; es—embryo sac; v—ovary vessel.

**Figure 10 plants-12-01038-f010:**
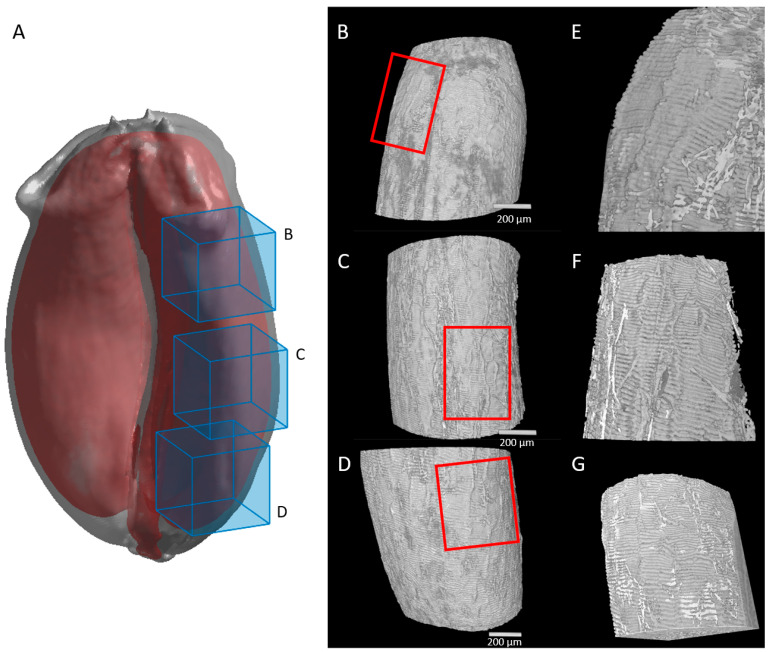
Morphology of cross cells at the surface of a caryopsis endocarp at 250 °DAA. (**A**) Localization of observed regions. Gray surface corresponds to the caryopsis outer surface, red surface corresponds to endocarp. (**B**–**D**) Reconstruction of endocarp surface showing cross cells at top (**B**), middle (**C**), and bottom (**D**) regions. (**E**–**G**) Zoom on the cross cells of the caryopsis in regions depicted by the red frames, highlighting the organization in patches.

**Figure 11 plants-12-01038-f011:**
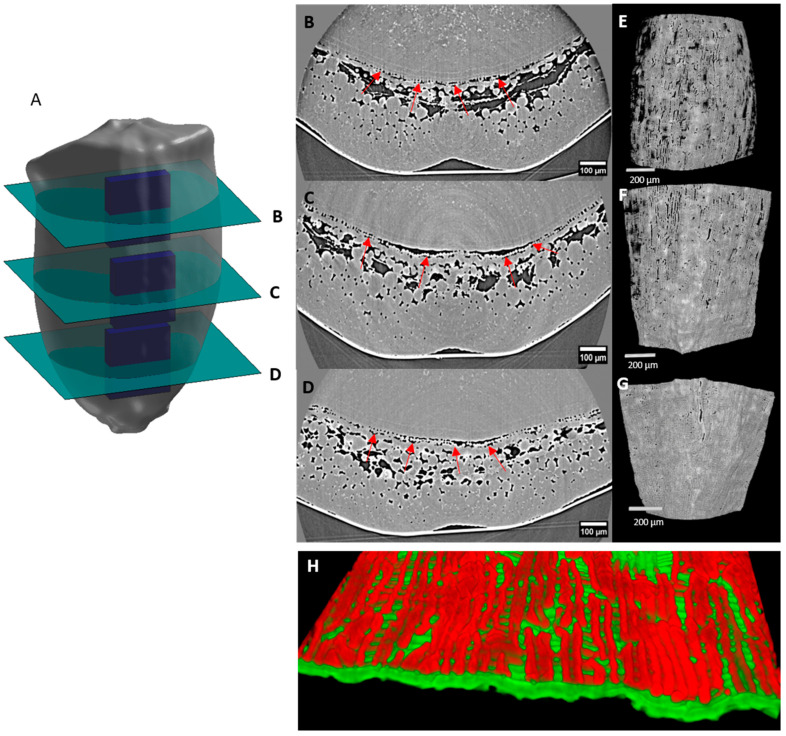
Longitudinal variability of the tube cells within the dorsal part of caryopses at 150 °DAA at different Z-positions. (**A**) Localization of the regions of interest: top (**B**,**E**), middle (**C**,**F**), bottom (**D**,**G**), using high-resolution tomography images. Red arrows on virtual X-Y sections (**B**–**D**) indicate the tube cells. (**E**–**G**) A 2D projection of a 3D visualization of the grid formed by the tube cells. (**H**) Example of superimposition of cross cells (green layer) and tube cells (red layer) in the dorsal × middle part of the wheat caryopsis.

**Figure 12 plants-12-01038-f012:**
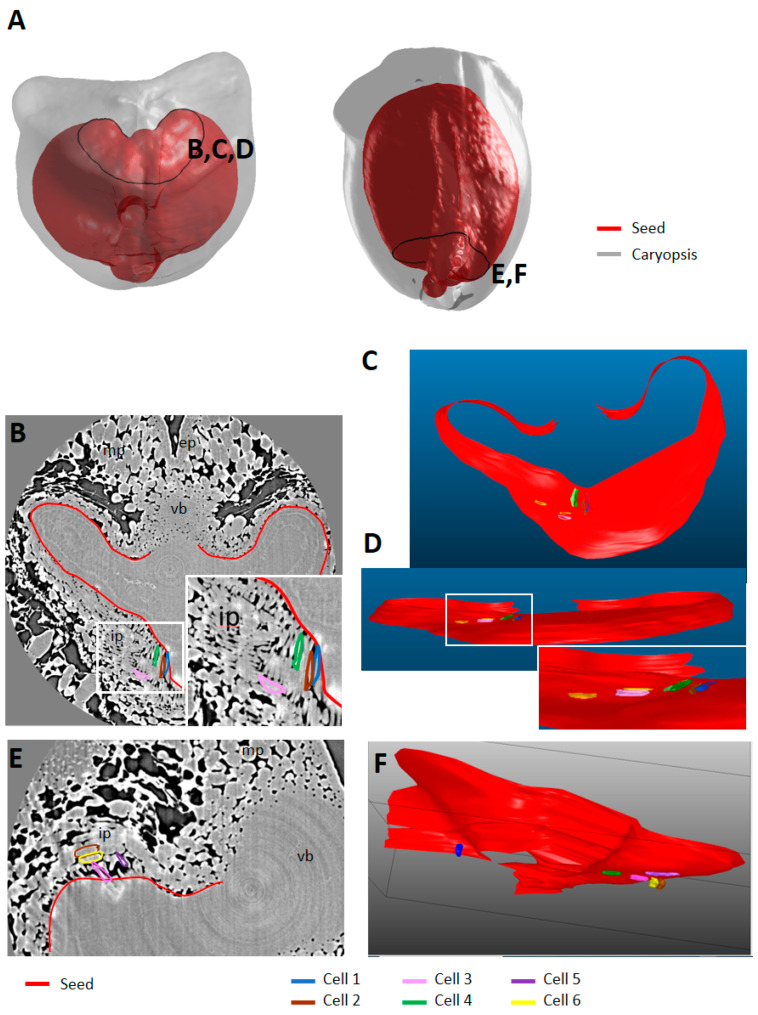
Endocarp cell orientation at the top and bottom of wheat caryopsis. (**A**) Localization of the region of interests (stage 150 °DAA). (**B**,**E**) Representative high-resolution images at the top (**B**) (zoom in the square to better visualize the delineated cells) and bottom of the caryopsis (**E**) (150 °DAA). (**C**,**D**,**F**) 3D volume reconstructions of part of the seed and endocarp cells at the top (top view (**C**); front view dorsal region (**D**) with zoom); and bottom of a representative caryopsis (**F**) (150 °DAA). The seed and several mesocarp cells were segmented and their volume was reconstructed using the Free-D software. Legend: mp—mesocarp; ip—endocarp; vb—vascular bundle; ep—epicarp.

**Figure 13 plants-12-01038-f013:**
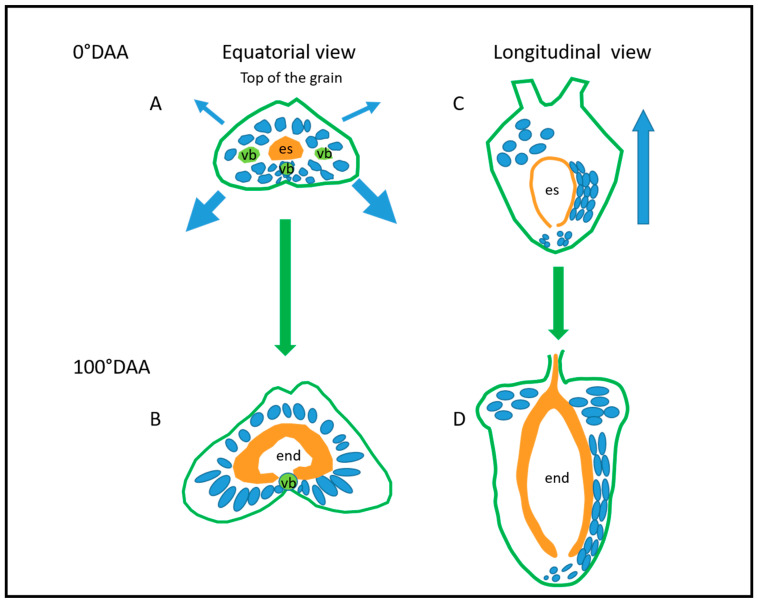
Schematic model of the main elongation axis of mesocarp cells related to the caryopsis growth. Mesocarp cells are represented in blue. This role is illustrated between the development stages 0 (**A**,**C**) and 100 °DAA (**B**,**D**) in two views: (**A**,**B**) equatorial and (**C**,**D**) longitudinal. Blue arrows indicate the main direction of caryopsis growth giving the orientation of mesocarp cells. Legend: es—embryo sac; end—endosperm; vb—vascular bundle.

**Figure 14 plants-12-01038-f014:**
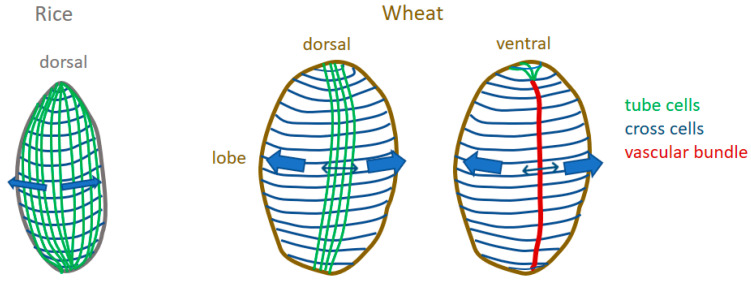
Schematic model of the putative function of endocarp in constraining growth. Within the rice caryopsis, cross cells and tube cells form a grid that covers the seed and would result in a uniform mechanical constraint. Within the wheat caryopsis, cross cells cover the whole surface of the seed while tube cells (and therefore the grid) are present only in the dorsal region. This could create local differences in mechanical properties, stiffer regions in the dorsal region and in the ventral region due to the vascular bundle, and “more” extensible regions in the lobes which could expand more.

**Table 1 plants-12-01038-t001:** Number of replicates for each development stage, expressed in thermal time after anthesis (°DAA) or in approximate number of days after anthesis (DAA).

Stage (°DAA)	0	25	50	80	100	150	180	200	250
Approximate stage (DAA)	0	1.5	3	5	6	9	11	12.5	15.5
Low Resolution	5	5	5	5	9	10	4	5	5
High Resolution	2		2		2	2		2	2

## Data Availability

The data presented here are available on request. The methodological developments are available either on request or directly from the links indicated within the manuscript.

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
