# Peer review of "New Growth-Related Features of Wheat Grain Pericarp Revealed by Synchrotron-Based X-ray Micro-Tomography and 3D Reconstruction"

_plants, 2023, doi:10.3390/plants12051038_

Round 1

Reviewer 1 Report

Review

New growth-related features of wheat grain pericarp revealed by synchrotron based X-ray micro-tomography and 3D reconstruction

The authors studied the 3D-anatomy of the growing wheat grain by X-ray micro-tomography. It is difficult to follow their claims due to lack of enough explanation of the figures. It is not clear to understand their new findings in this manuscript.

p3 L107, p13 L284: "3" is missing.

p19 L439: For this section, the authors should provide a schematic model like Fig.12.

p21 L512: The authors should clearly mention their new findings in this study.

p23 L634: Font size should be adjusted.

Fig.2: A seed image of 250°DAA should be provided.

Fig.9 legend: The size of scale bars should be mentioned.

Fig.11: It is very difficult to understand. The authors should explain colored lines in B and E, and painted areas in C, D and F. In this figure, they named red part as "seed", but they named the same regions as "inner regions" in Fig.1. The term of seed should be used for the whole part of the grain. They might name it as "endosperm" even in this stage as ref.24. The uses of these terms should be consistent with those in Fig.12.

Figs: The position of aleurone layers should be indicated.

Reviewer 2 Report

Journal Plants (ISSN 2223-7747)

Manuscript ID plants-2115651

Type Article

Title New growth-related features of wheat grain pericarp revealed by synchrotron-based X-ray micro-tomography and 3D reconstruction

Authors David Legland , Thang Duong Quoc Le , Camille Alvarado, Christine Girousse, Anne-Laure Chateigner-Boutin

Section Plant Development and Morphogenesis

Collection Feature Papers in Plant Development and Morphogenesis

Dear Editor,

Manuscript ID plants-2115651 requires profound revision and reediting.

Please find my comments below.

Best regards

Introduction

1. I suggest organizing the Introduction text in accordance with the chronological subsections following one after the other.

2. Please add information about the aleurone layer and the type of proteins very important from the nutritional point of view dominating in the chemical composition of wheat grains.

3. Similarly, add information about other biologically active chemical compounds and their pro-health role.

4. Indicate the type of starch that predominates in wheat grains.

5. Indicated the group of readers who may find the research useful.

6. Give a detailed rationale behind the undertaken study.

7. Specify the research theses.

8. Formulate the specific aim of the study.

Results

9. l. 98-106 Please eliminate the information that should be included in the Material and methods subsection (similarly in the entire subsection).

10. Figures 1-11. The numerical values of the bars should be placed on individual photographs (a specific value above a given bar). The reader expects more information in the photo description (please provide specific information on what the authors wanted to present in the photos). A reader with no experience with anatomy (fruit and seed coat, aleurone layer, endosperm, embryo, furrow) should be provided with information immediately while looking at the photographs.

Discussion

11. l. 341 correct “([17,24] etc).”

12. l 305 correct “(Figure 10D,E,F, G)” – this should be “(Figure 10D-G)” throughout the text

13. Eliminate the redundant space in (Figure 11 B), (Figure 11 C, D)

14 l. 385 – 400, 417 – 431, 472 – 476, 484 – 490 etc. Eliminate citation of the same reference (twice) here and throughout the text.

15. l. Some fragments should be reedited to have a form of a discussion instead of a literature review.

16. Indicate the research perspectives in the future.

Material and methods

17. l. 520 – 530  Eliminate the repetition of the phrase “…as described in [21].”

18. Complete the microscopy methodology, preparation of semi-thin microscopic sections, e.g. fixing, dewatering, embedding, cutting (what they were cut with), type of microtome, section thickness, etc.

19. Complete the number of fragments taken and observations made to eliminate accidental sampling.

Conclusions

20. Please formulate specific conclusions from your research.

References

21. The list of references should be screened in detail step by step and corrected in accordance with the guidelines for authors.

Reviewer 3 Report

In their manuscript, Legland et al. has presented an interesting and original work which shows the possibility of X-ray micro-tomography for investigations on plant tissues and cells using the pericarp of the developing wheat grain as the example. In the present form, the manuscript, however, still requires extensive improvements. I have tried to go through the text and highlight the most important positions to correct. But I have to stress that these propositions are not exhaustive.

 Major.

1.       I strongly recommend to redraw figures 2, 5, 7, 8, 9, 10 and 11. The figure should show on the introductory grain photos, schemes or cartoons where exactly the sections were virtually of physically performed and describe all important tissues. This will tremendously help to follow the manuscript. I have seen thousands of figures of virtually of physically dissected cereal grains before. But here, I had problem in many cases to correctly imagine where the particular virtual section took place. It will be not possible to understand the figures for readers not familiar with the seed structure. As the example, the positions for virtual sections is shown at Fig. 2 by blocks in (A), but these blocks correspond only to the sections shown in (D). The figures (C) and (D) show sub cut-outs of some sections. From which part of a seed exactly are they coming? By the way, should be cutting positions in (A) really visualized by blocks or better lines? Similar descriptions for positions of the sections should be done for all relevant figures

2.       The scientific terms are often used inappropriately.

(i)  Especially the term “seed” (as examples, lines 222, 440 and all others) is used totally wrong way. Instead, the term “endosperm” must be applied in all these cases (the most inner seed organ covered by the number of maternal tissues including pericarp). To clarify the terms:

seed (present in all angiosperms, the broad term meaning the entire organ for reproduction) = grain (only in grasses, more specific) = caryopsis (botanical term for a Poaceae seed).

(ii)  the authors have used several different terms for the most outer pericarp layer (e.g., epidermis, epiderm, epicarp). They mean the same tissue in a cereal grain. Please stick to one of them only.

(iii) The term “slice” should be “section” or “virtual section”.

3.       The description of grain development (lines 43-45) is not correct. The developing embryo and endosperm are covered by a number of maternal tissues (nucellus, nucellar epidermis, testa (integuments) and pericarp). The pericarp represents a bulk of the maternal tissues. Please rewrite correctly.

4.       The authors have applied the quite unusual system for staging of samples (degree-days after anthesis). Most of research use the simplified system as days after anthesis (DAA). Clear, the grain development depends on the temperature. However, to understand the staging correctly by other researchers, the authors should provide the temperature regime used in their green house (in Material and methods section) allowing the reader to recalculate the staging. Additionally, I would recommend to add DAA data to the Table 1.         

5.       English must be improved in both grammar and style. There is impression that the Introduction and Result sections were written by different people. The terms, correctly used in the Introduction, were not more correct in the Results.  

More specific comments:

Line 51. What does it mean here and below “outer tissues”?

L58-59. The sentence is not correct. Water has much more functions in the developing seed.

L73-74. The sentence is not correct: the most of pericarp cells completed their proliferation at the anthesis.

L77. The sentence is not fully correct. One of the most important functions of the pericarp is serving as transient storage tissue.

L131. Please specify the tissue in the sentence “High-resolution images revealed individual cells in several grain tissues”.

L134. “In contrast, cells inside the embryo sac at anthesis or later in the seed tissues are difficult to visually separate as there is no void between them (Figure 3).” Figure 3 does not show embryo sac at anthesis. Please correct.

L197 (and further figure legends). The sentence “3D view of the grain surface showing stomata in the ventral region” is wrong. The figure(s) shows 2D projections of the 3D view. Please correct.

L227. “…mesocarp tissue contains large voids surrounding the seed due to developmental mesocarp lysis,” is not correct. Mesocarp undergoes programmed cell death or disintegration.

L284. What does it mean “D reconstruction”?

L338. “with different functions…, all driven to the production of a new seedling.” Not all seed functions are devoted to production of a seedling (some deals with seed protection, as example). Please rewrite.

L373. “mesocarp cells are parenchyma cells”. Strongly speaking, parenchyma cells are present only in leaves. Here is more appropriate to call “parenchyma-like cells”.

L374-376. The sentence here has no sense. Please rewrite.

L453. The sentence “The grain contains several epidermal tissues (epicarp, inner pericarp, testa and nucellar epidermis)” is not correct. The grain has only one epidermal tissue, namely epidermis (or epicarp). All above mentioned tissues are maternal tissues (all these tissues are derived from mother plant and not from fertilization events).

L466. The subtitle “3.2.3. Role in Photosynthesis” is fully inappropriate. The authors have not analyzed the grain photosynthesis at all and therefore cannot discuss this. But they can discuss with good grace the role of stomata and void spaces, which they have (re-)discovered in the work.

This is not exhausting list of wrong or mistaken positions in the text. Nevertheless, the manuscript may be of interest for researchers dealing with the seed development. The manuscript requires deep improvement in description of results and discussion as well as in English grammar.

Round 2

Reviewer 1 Report

In the legend of Figs.2, 3, 4, 9, and 12, Mp should be mp.

In the legend of Fig.11, the last line should be (H).

Reviewer 2 Report

Dear Editor

The revised version of the manuscript does not include all the changes that are necessary, especially for an article published by an international journal.

A.    All presented photographs (marked with numbers) in a given figure should be cited, e.g.:

1. Figure 1A-I: The authors cite only photographs 1A and 1B, whereas the other photographs 1C-I are not cited.

2. Figure 3A-C: no citation of 3A and 3B.

3. Figure 4: no citation of the individual photographs 4A, 4B, and 4C.

4. Figure 5: no citation of 5E.

5. The citations of all photographs in figures 6-12 (indicated by the letters in the given figures) should be completed.

B.     The adjacent citations of the same literature reference have not been eliminated.

6. In the discussion subsection, 23 literature references are cited, but this number includes repetitive citations, e.g. reference [17] is cited as many as 6 times, reference [20] x 2, [21] x 3, [34] x 3, [37] x 2, and [38] x 2.

7. In subsection 3.1, reference [17] is cited three times, including twice in direct succession; similarly, reference [34] is quoted in the immediate vicinity of each other.

8. This comment applies to reference [38] in subsection 3.2.1.

9. Discussion is not an appropriate place to refer to the figures (5 figures are cited, including twice citations of figure 14).

C.    There are still errors in the “References” subsection; the text does not comply with the guidelines for authors. I have marked some errors in the PDF text.

In this review a review, I have indicated minor corrections, but I leave the decision on further processing to the Editor. The publishing rules and editorial requirements cannot be ignored.

Some comments are highlighted in PDF file.

Sincerely,

Reviewer

Reviewer 3 Report

The authors significantly improved the quality of the manuscript. Now it is easy to follow and clear to understand. The significance of the investigation is better highlighted. Most of the figures are of good quality (but see below). I also now agree with the usage of the scientific terminology. However, to my opinion, English still requires further improvements, especially in the Discussion part. Please use shorter sentences. It will allow more consequent grammar usage and better controlling of sense. Below I have provided some points which need to be improved. But they are not exhausting.

Figure 14 has to be improved. Please take away the word underlining at the figure.

Other remarks.

Please avoid ellipsis (…) throughout the text. Please write exactly in each case what do you mean.

Lines 59-60. please check the grammar in the sentence starting with “First because at early stage…”

L. 72. please check the grammar in the sentence starting with: “The endocarp differentiates…”

L. 257: take away a dot (.).

L. 530 “3.2.3. Putative role in caryopsis photosynthesis” Of what?

L. 547. Please replace the word “nutrients” by “assimilates”. “Nutrients” is a broader term and not all kinds of nutrients are provided by photosynthesis.

L. 554. Please re-check grammar of this sentence.

L. 588. Please change from “a first step could the constitution of a 3D atlas of the whole caryopsis from whole caryopsis images, by…” to “the first step could be the constitution of a 3D atlas of the whole caryopsis images by…”

L. 592. Please replace “physics” by “physical”

L. 702. Please replace “4.2.73. D visualization of..” by “4.2.73. 3D visualization of..”

L. 717-719. Please check the grammar of the sentence.
